# Research on Micro-Quantitative Detection Technology of Simulated Waterbody COD Based on the Ozone Chemiluminescence Method

**Peng Li [1], Yizhuo Wang [1,2] and Bo Xu [1,\*]**

[1] Mechanical Engineering College, Yanshan University, Qinhuangdao 066004, China; lemmber@163.com (P.L.); wangyz0531@163.com (Y.W.)

[2] Mechanical Engineering College, Beihua University, Jilin 132022, China

[\*] Correspondence: xubo0609@163.com

**Abstract:** Chemical oxygen demand (COD), reflecting the degree of waterbody contaminated by reduction substances, is an important parameter for water quality monitoring. The existing measurement method of waterbody COD takes time and is a complex system, which cannot meet the real-time monitoring requirements of river pollution indicators. We developed the vortex t-structure microfluidic detection chip with the help of microfluidic technology and designed the COD detection system with a high integration degree based on the principle of ozone chemiluminescence, and we have also carried out research on a waterbody COD quantitative detection test. The test results show that the detection chip can generate quantitative and controllable ozone-based bubbles; it also shows the advantages of a simple system and short test time without environmental pollution, which provides some technical support for the online real-time monitoring of river water quality.

**Keywords:** ozone base micro-fine bubbles; chemiluminescence method; microfluidic chip; COD detection

## 1. Introduction

Chemical oxygen demand (COD) refers to the oxidation dose consumed when water samples are treated with strong oxidants under certain conditions, which is used as an international standard index to evaluate the degree of organic pollution [1,2]. The traditional COD analysis method is to use dichromate or potassium permanganate and other strong oxidation reagents to make organic matters in water or wastewater degrade and oxidize, and these reagents have been established and standardized in many countries. However, this method is not environmentally friendly [3,4], is inconvenient to operate [5], and is difficult to carry out large-scale measurements quickly (in 2–4 h) [6].

In recent years, rapid, sensitive and eco-friendly COD determination methods have been developed. At present, there are new COD determination methods reported, including the electrochemical method [7–10], photocatalytic method [11], direct spectroscopy method [12] and fluorescence detection method [13]. Compared with traditional methods, these methods are simple, fast and sensitive. However, for electrochemical methods, many electrodes developed in the past, such as $PbO_2$, AgO/CuO, Cu/CuO or diamond, nanocopper grade electrodes, either have the potential to release toxic metal ions (such as lead) or they are expensive to manufacture and use. As for photocatalytic method, due to the limited light energy conversion of semiconductor materials, the oxidation capacity of organic matter in water samples is not high, and most $TiO_2$ based methods have some shortcomings, such as limited amount of dissolved oxygen in the sample and narrow analysis range. In addition, for optical detection methods, the cost is high, and it usually requires large and expensive instruments and is difficult to apply to field monitoring.

It is a relatively novel online monitoring technology—ozone chemiluminescence determination of COD in the waterbody. In other words, ozone dissolves in water and

decomposes, resulting in a series of strong oxidizing active free radicals. The organic matter oxidizes in water and the characteristic of chemiluminescence is generated [14] to realize the determination of COD in the waterbody. However, ozone itself is difficult to dissolve in water and is easy to decompose into oxygen at room temperature, and excessive ozone is a toxic gas. Therefore, the premise of using ozone chemiluminescence to detect COD value of water involves improving the utilization rate of ozone and realizing the quantitative and controllable generation of ozone. Microfluidic technology can use microfluidic chips to prepare microbubbles with controllable volume and generation frequency, and the device has a simple structure with low power consumption [15–17]. In addition, the generated microbubbles have the advantages of a large specific surface area, small rise rate [18,19] and high mass transfer efficiency [20]. Therefore, the introduction of microfluidic technology combined with the principle of ozone luminescence can realize the detection of COD in water [21], with the advantages of high sensitivity, fast processing, and no secondary pollution.

In this study, a simple, low-cost, rapid and secondary contamination-free method for COD detection in water samples was successfully established by combining ozone chemiluminescence with microfluidic technology. In order to achieve the quantitative and controllable generation of ozone-based bubbles, this paper first carried out an experimental study on the bubble formation characteristics of a T-type microchannel and verified that the T-type microchannel structure has certain advantages in the formation of bubbles. Second, a vortex T-type microfluidic detection chip was developed to evenly detect the luminescent signal. Finally, combined with the principle of ozone chemiluminescence, a highly integrated COD detection system was built, and the detection test of simulated water COD (glucose solution) was completed. The test results showed that the measured values of the system are consistent with those of the traditional method, and it has the advantages of a simple system, short test time, and environmental protection without pollution, which all provide a certain technical support for the on-line real-time monitoring of water quality.

## 2. Experimental

### 2.1. Detection Principles

Organic compounds in sewage generally include nitrogen compounds (NOx), sulfur compounds (Sox), hydrocarbons and so on, which can produce redox reactions with ozone to produce chemiluminescence [22,23]. When different organic compounds react with ozone, it is accompanied by chemiluminescence of different light intensities, and waterbody COD values can be obtained indirectly according to the correlation between luminescent intensity and waterbody COD [24]. Its detection principle is shown in Figure 1:

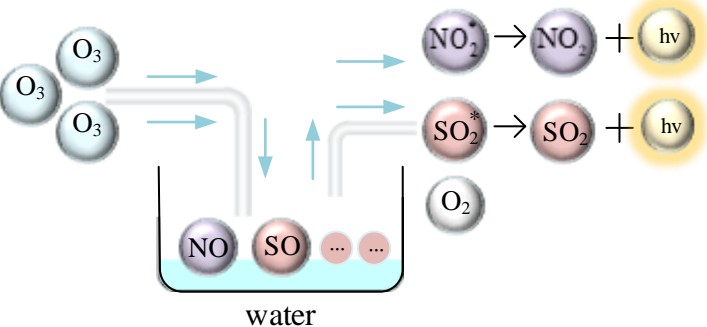

**Figure 1.** Ozone chemiluminescence COD detection principle diagram.

### 2.2. Detection System

We used the microfluidic chip as the test carrier to inject a certain concentration of ozone and organic solution into the microchannel for the hybrid reaction. We also used the microlight detection device to detect the produced chemiluminescence during the reaction,

which linearly fit the average peak of the electrical signal to the COD value detected by potassium dichromate after the conversion of the photoelectric signal. The COD of the organic solution to be tested was detected indirectly by using the fitting correspondence between the two. The ozone chemiluminescence COD detection test process is shown in Figure 2.

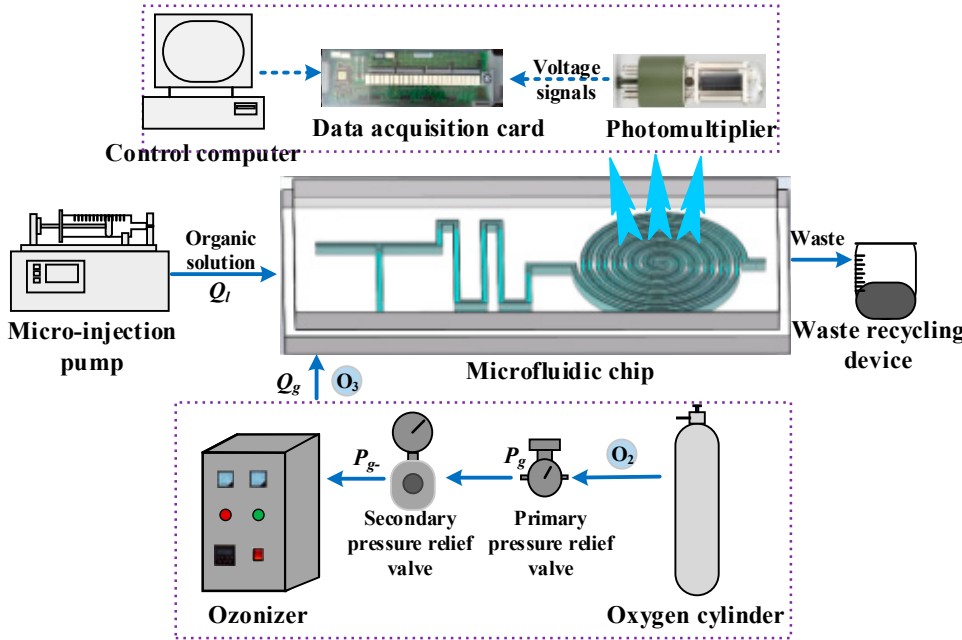

**Figure 2.** Flow chart of the ozone chemiluminescence COD detection system.

The COD detection test system was constructed according to the flow chart of the ozone chemiluminescence COD detection test, as shown in Figure 3.

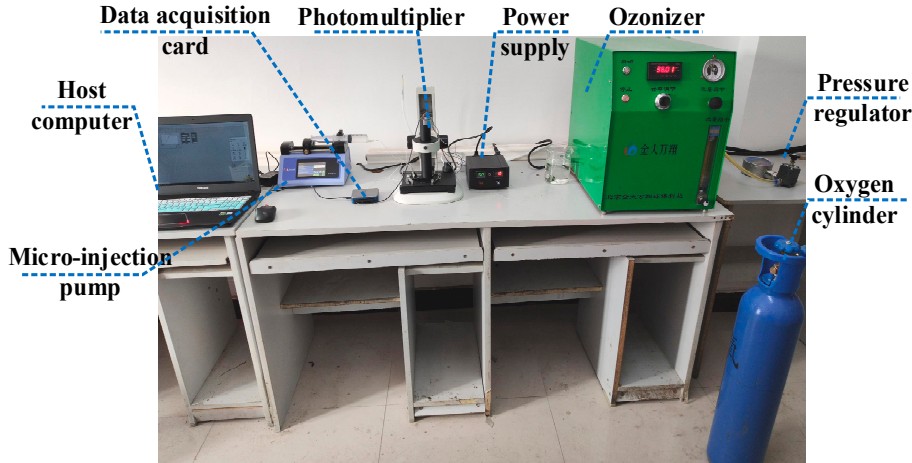

**Figure 3.** Physical diagram of the ozone chemiluminescence COD detection system.

### 2.3. COD Detection Chip Design

In order to fully mix the ozone with the glucose solution in the microchannel, to undergo chemiluminescence in a fixed area and to enable one to uniformly detect its produced chemiluminescence by the optomultiplier tube, we developed a microfluidic chip [25] with both a T-type and vortex structure, as shown in Figure 4. The width of all microchannels in the chip is 0.5 mm, and the depth is 0.25 mm. The liquid and gas channels

adopt a T-type flow focusing structure, and the size of the elliptic long and short axes in the vortex region are 2.2 and 1.3 mm, respectively.

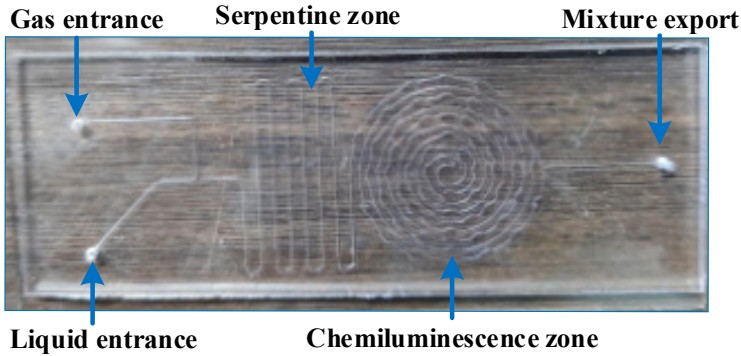

**Figure 4.** Vortex-type T-type structure microfluidic chip.

Because there is a certain gap between the producing ozone base bubbles and the liquid phase, we set four sets of "serpentine" microchannels at the end of the T-type microchannel to achieve the full pre-mixing of gas-liquid two-phase fluids; the vortex microchannel region is the core area of ozone chemiluminescence, and the aim of using the vortex-type structure is to make the chemiluminescence evenly diverge; there are several elliptical grooves evenly distributed in the vortex region to achieve the short residence mixing and full reaction of the gas–liquid two-phase fluid to enhance the chemiluminescence intensity.

By using a vortex-type T-type structure microfluidic chip to perform ozone chemiluminescence testing, the chemiluminescence sinusoidal signal can be detected under appropriate liquid flow and gas pressure ratio conditions. We selected a specific working condition of liquid flow rate 4 mL/h and gas pressure 50 kPa, and the chemiluminescent sinusoidal voltage signal detected by the photomultiplier is shown in Figure 5.

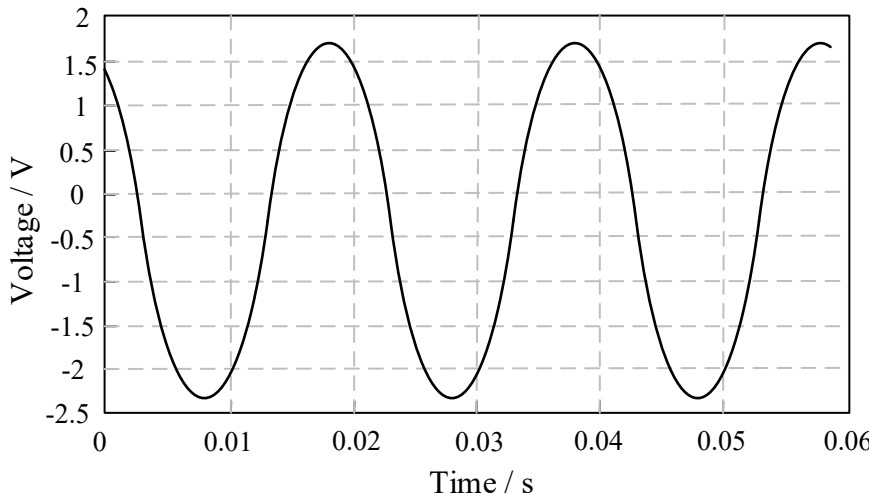

**Figure 5.** Ozone chemiluminescence sinusoidal voltage signal.

## 3. Results and Discussion

### 3.1. Bubble Formation Characteristic Test in T-Type Channel

T-type structural microchannels are the typical channel configuration, where gas and liquid flow through two microchannels and meet at T-type junctions, and where gas will form micro-fine bubbles, whose diameter is smaller than the cross section size of microchannel under the action of liquid flow shear. Compared with other configurations, the T-type microchannel configuration is simple and easy to realize the control of the size and formation frequency of micro-fine bubbles.

In order to explore the formation characteristics of micro-fine bubbles under T-type flow-focus, we selected polydimethylsiloxane (PDMS) as the microfluidic chip material. We developed a T-type flow-focused microfluidic chip according to the structure and distribution of the T-type flow-focused gas-liquid two-phase microchannel. The shape structure and size of the chip are shown in Figure 6a, and the test system is shown in Figure 6b.

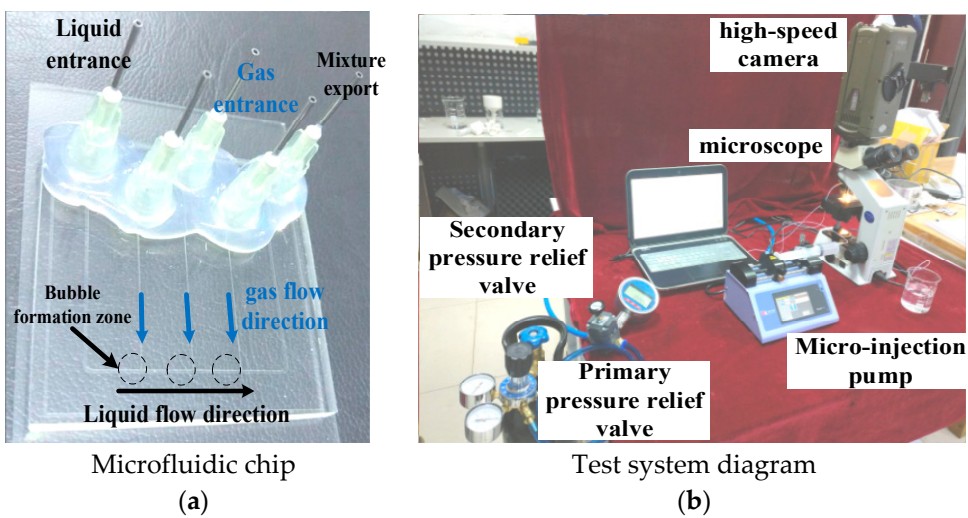

Microfluidic chip

(**a**)

Test system diagram

(**b**)

**Figure 6.** Test environment for micro-fine bubble formation characteristics.

(1) Bubble Formation Time Test. During the test process of exploring the formation time of micro-fine bubbles, we fixed the gas channel width to 0.15 mm, and selected three gas pressure regulation gradients at 60, 65 and 70 kPa. We adjusted the liquid flow of the equal gradient rate under any gas pressure and explored the bubble formation time under different liquid flows and gas pressure. The change laws of bubble formation time under different working conditions is shown in Figure 7.

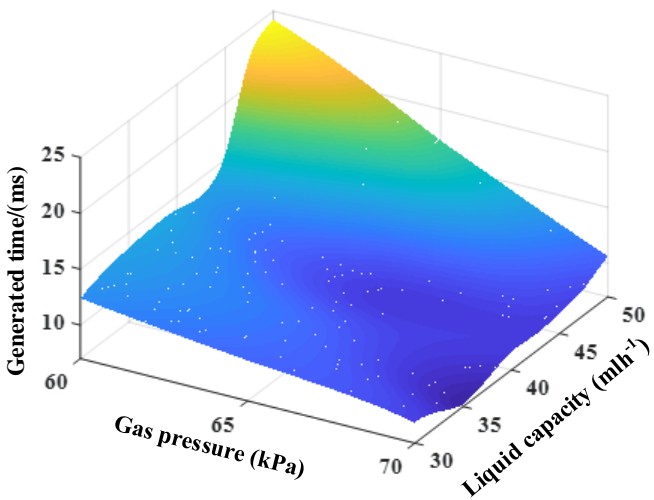

**Figure 7.** Micro-fine bubble formation time under different liquid flows and gas pressure.

In Figure 7, when the pressure value of each liquid is fixed, the formation time of the micro-fine bubbles increases if the liquid flows with a trend of dramatic increase. Conversely, the bubble formation time all decreases with the increase in gas pressure.

(2) Bubble Disengagement Volume Test. During the process of exploring the test of micro-fine bubble disengagement volume, the fixed gas channel width was 0.15 mm. The test was conducted according to the adjustment range of the liquid flow and gas pressure

in the test planning, and finally, we obtained the change law of the micro-fine bubble disengagement volume along with the liquid flow and gas pressure as shown in Figure 8.

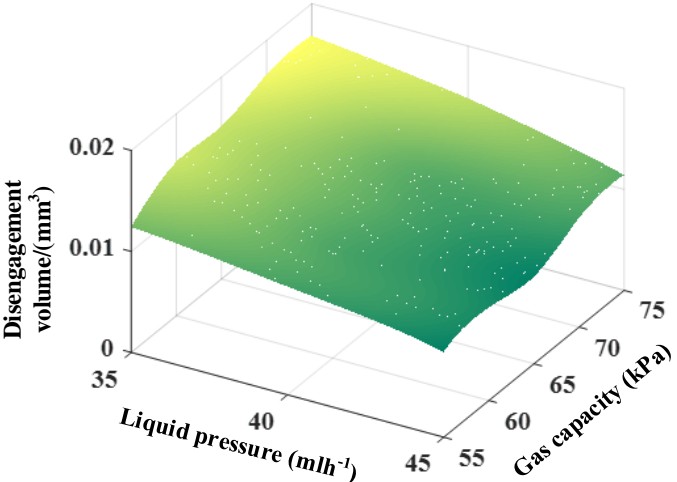

**Figure 8.** Micro-fine bubble disengagement volume under different liquid flows and gas pressure.

It is easy to find, by analyzing Figure 8, that when the gas pressure is constant, the bubble disengagement volume shows the trend of an approximately linear decreasing with the increase in the liquid flows. Conversely, when the liquid flows is fixed, the bubble disengagement volume increases with the increase in the gas pressure and the trend to increase is non-linear.

Through the above test, we further verified that the T-type structural microchannel has some advantages for the formation of micro-fine bubble. Moreover, the microfluidic chip based on PDMS material can basically achieve the stable control of bubble volume and formation frequency, which provides a reference value for the subsequent microchannel structure design of an ozone chemiluminescence-based COD microfluidic detection chip.

In the subsequent COD test process, if the ozone amount is insufficient, the organic matter cannot be fully oxidized to produce chemiluminescence; and if the ozone is excessive, it is easy to leak and pollute the environment. Therefore, the COD detection chip can also be designed as a T-type structure microchannel and can form the ozone base bubbles through flowing shear. In addition, wrapping the ozone with bubbles can effectively control the ozone equivalent per unit of time and can realize the full and effective reaction of ozone and organic matter to produce the best intensity of chemiluminescence.

### 3.2. COD Calibration

Before COD measurement, the glucose solution with fixed concentration gradient needs to be COD calibrated by the standard method. Since glucose solution in simulated water is used in this test, there will be no interference of BOD. According to *the Environmental Quality Standard of Surface water of the People's Republic of China*, the surface water quality in China can be divided into Classes I, II, III, IV, V, and inferior Class V, and the COD indexes of various water quality tolerance are specified as shown in Table 1.

According to the COD values of various water qualities in the table, the corresponding relationship between glucose quality and COD is obtained through the standard chemical equation of glucose oxygen consumption. Namely, 1 g glucose corresponds to 1.067 g cod, in order to obtain the quality of glucose required to meet the COD indexes of each water quality. Then, 10 groups of fixed concentration glucose solutions were prepared, and the COD value of each concentration glucose solution was detected by potassium dichromate standard method. Some test results are shown in Figure 9, and the detailed glucose solution concentration and COD test data of standard method are recorded in Table 1.

**Table 1.** Corresponding relationship between surface water quality standard tolerance COD and glucose standard detection COD.

| Water Quality Classification (mg/L) | Class I | Class II | Class II | Class III | Class IV | Class IV | Class V | Class V | Inferior Class V | Inferior Class V |
|---|---|---|---|---|---|---|---|---|---|---|
| Tolerance COD | 5 | 10 | 15 | 20 | 25 | 30 | 35 | 40 | 45 | 50 |
| Glucose Concentration | 4.69 | 9.37 | 14.06 | 18.74 | 23.43 | 28.12 | 32.8 | 37.49 | 42.17 | 46.86 |
| Detection of COD by Potassium Dichromate Method | 4.3 | 10.2 | 15.3 | 19.7 | 25.5 | 31.2 | 35.5 | 39.8 | 45.1 | 50.8 |

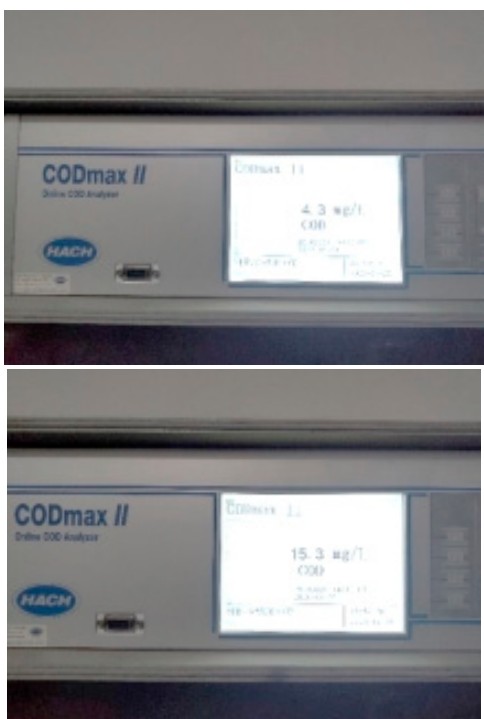
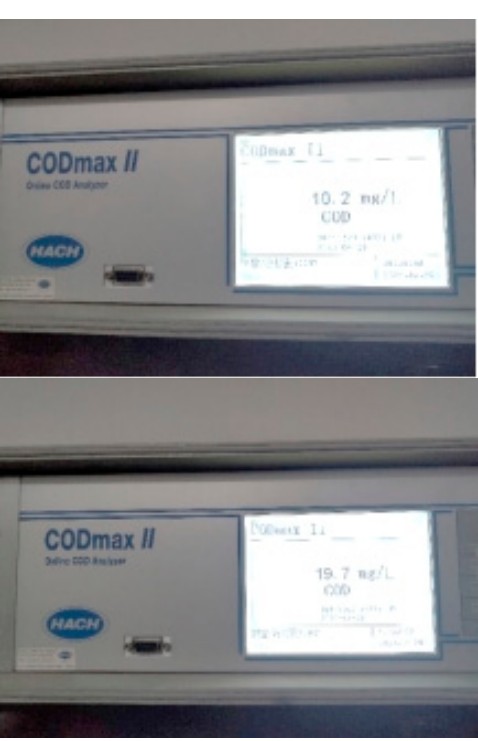

**Figure 9.** Measured data of COD in glucose solution detected by potassium dichromate method.

*3.3. Optimal Gas–Liquid Flow Selection*

In addition, before carrying out a COD detection test, it is necessary to find the ratio group that can obtain the maximum average peak value of voltage waveform from the array capacity and pressure ratio and take this group of working conditions as the standard to detect COD.

In order to obtain the waveform signal that can significantly produce large chemiluminescence voltage, 150 mg/L glucose solution is configured as the reagent to be tested, and the optimal gas production power of ozonator is 600 W; second, the ozone chemiluminescence COD detection between 10 liquid flows with different gas pressures and fixed gradients was carried out by using the control variable method, fixing the gas pressure of 40~70 kpa and changing the liquid flow of 1~10 mL/h.

After obtaining the waveform of chemiluminescence voltage conversion sinusoidal signal under different working conditions, the discrete data of voltage sinusoidal waveform was collected by LabVIEW data acquisition software; then the positive value data of all voltage peaks under each working condition for statistical sorting was selected, which enabled the obtainment of the average peak value of ozone chemiluminescence signal under different working conditions, as shown in Table 2.

**Table 2.** Average peak value of chemiluminescence voltage signal under different liquid capacity and gas pressure (V).

| Liquid Capacity (mL/h) | | 1 | 2 | 3 | 4 | 5 | 6 | 7 | 8 | 9 | 10 |
|---|---|---|---|---|---|---|---|---|---|---|---|
| | 40 | 1.61 | 1.03 | 1.68 | 1.71 | 3.09 | 1.23 | 1.31 | 1.09 | 0.52 | 0.45 |
| | 45 | 1.38 | 0.64 | 1.68 | 1.56 | 1.15 | 1.99 | 0.89 | 1.85 | 0.68 | 0.57 |
| | 50 | 0.31 | 0.31 | 2.31 | 3.68 | 1.77 | 1.62 | 1.58 | 1.84 | 0.82 | 0.74 |
| Gas Pressure (kPa) | 55 | 1.35 | 1.76 | 0.45 | 1.86 | 2.27 | 0.24 | 1.02 | 1.66 | 1.6 | 1.35 |
| | 60 | 0.46 | 0.9 | 1.57 | 1.32 | 1.09 | 1.15 | 0.97 | 2.14 | 1.34 | 1.25 |
| | 65 | 0.37 | 0.48 | 0.123 | 0.63 | 1.02 | 1.87 | 0.58 | 1.31 | 0.74 | 0.69 |
| | 70 | 0.45 | 0.09 | 0.51 | 0.56 | 1.41 | 1.88 | 0.92 | 0.54 | 0.5 | 0.23 |

As can be seen from Table 2, for the 150 mg/L glucose solution (a set of glucose solution concentrations is used to find the best gas–liquid flow), the average peak value range of voltage signal obtained under different working conditions is between 0~4 V. The data in the Table are further graphically processed to obtain the average peak value change law curve of chemiluminescence voltage waveform signal, as shown in Figure 10.

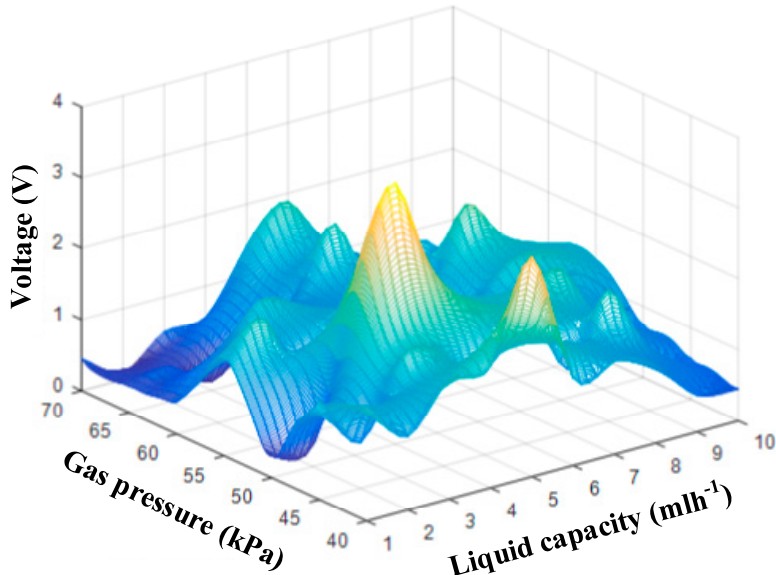

**Figure 10.** Law curve of average peak value change of chemiluminescence voltage waveform under different working conditions.

Figure 10 describes the change law of the average peak value of chemiluminescence voltage signal under different working conditions. Namely, the average peak value of voltage increases gradually from low and then decreases gradually. According to Table 2, when the liquid capacity is 4 mL/h and the gas pressure is 50 kPa, the maximum average peak value of the voltage signal is 3.68 V; ignoring the test system error and measurement error, the ozone chemiluminescence COD detection test is planned to be carried out under this working condition.

*3.4. Generation of Ozone Bubbles in Microchips*

Before the ozone chemiluminescence COD detection test was carried out, the formation of ozone-base micro-fine bubbles at the T-shaped structure in the microfluidic chip was observed by the microscope. The results are shown in Figure 11.

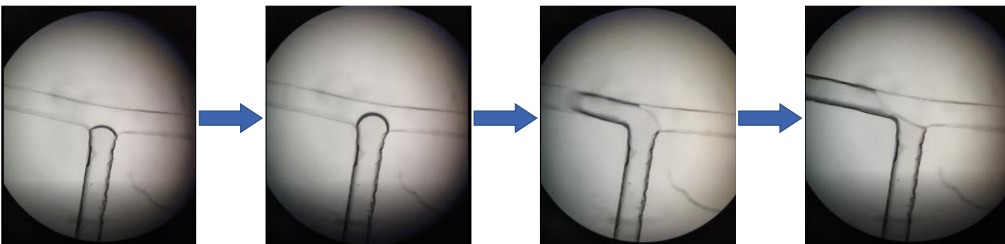

**Figure 11.** Timing chart of ozone column generation.

It can be seen from the figure that micro-fine bubbles with regular and clear shape could not be generated in the COD detection chip but could finally separate in the form of a "gas column". The reason may be that the glucose solution and ozone used in this test are prone to redox reaction; the concentration and viscosity of glucose will decrease after the reaction, and the microchannel width of the chip is relatively large; thus, regular bubbles cannot be generated.

Although the micro-fine bubbles with the volume of the micron cannot be generated in the COD microfluidic detection chip, ozone bubbles with a slightly larger volume are still generated based on the T-shaped microchannel structure; this phenomenon is conducive to the long-term retention of ozone in the microchannel and the full mixing reaction with glucose solution in the vortex region, in order to improve the chemiluminescence reaction efficiency and enhance the chemiluminescence intensity.

### 3.5. Test Verification

According to the specifications in Table 1, 10 groups of glucose solutions with the same concentration for later use were prepared, as shown in Figure 12. Then, the ozone chemiluminescence COD detection test system was built, and 6 groups of 10 groups of solutions were selected to carry out the ozone chemiluminescence COD detection test with the optimal liquid flow rate and gas pressure.

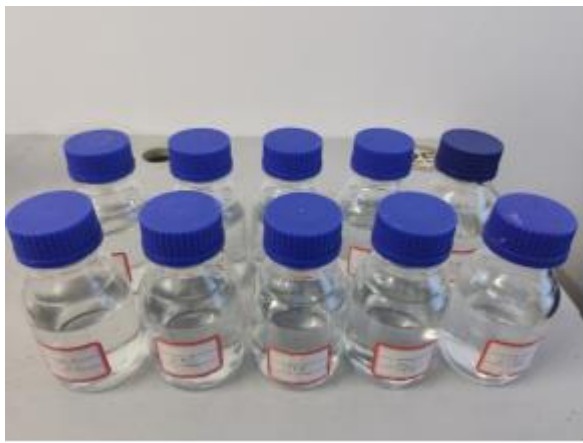

**Figure 12.** Ten groups of standard glucose solutions with fixed concentration.

In the test process, the output voltage waveform of ozone chemiluminescence was filtered by FIR low-pass digital filter at first, and the stable output sinusoidal voltage was obtained. Then, according to the voltage of each group of glucose solution, the output waveform was dynamically displayed for 10 s at the sampling rate of 1000 and the sampling number of 60. The discrete value of voltage sinusoidal signal corresponding to each group of solution was recorded and saved by LabVIEW file storage function. Finally, the average of all positive peaks of sinusoidal voltage discrete values of each group of solutions within 10 s was taken to obtain the average peak values of chemiluminescence signals of glucose

solutions at the corresponding concentration, and the average peak values of voltage of six of the groups of solutions were recorded in Table 3.

**Table 3.** Average peak values of sinusoidal voltage corresponding to six groups of glucose solutions and COD value detected by potassium dichromate.

| Glucose concentration (mg/L) | 4.69 | 14.06 | 18.74 | 23.43 | 37.49 | 42.17 |
|---|---|---|---|---|---|---|
| Voltage average peak values (V) | 1.55 | 1.62 | 1.64 | 1.67 | 1.76 | 1.79 |
| Determination of COD by potassium dichromate | 4.3 | 15.3 | 19.7 | 25.5 | 39.8 | 45.1 |

The average peak value of chemiluminescence signal of six groups of glucose solution in Table 3 linearly fits with the COD value detected by potassium dichromate method in the least square method. The fitting results are shown in Figure 13.

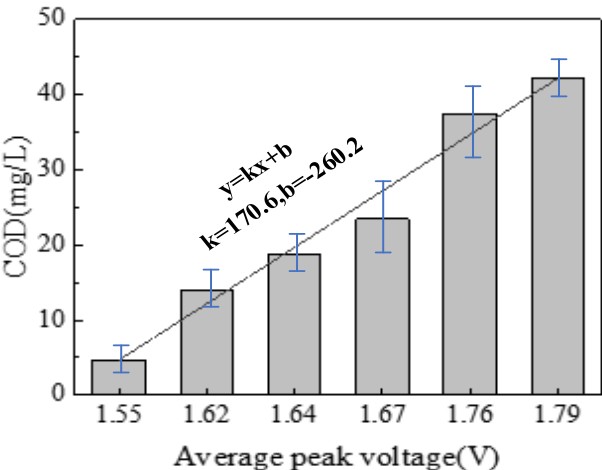

**Figure 13.** Linear fitting result of voltage average peak values and COD detection values by potassium dichromate method.

It can be seen from Figure 13 that the voltage average peak values corresponding to the six groups of standard glucose solutions have a good linear fitting effect with the detection of COD by the potassium dichromate method. The correlation between the COD values of glucose solution and the voltage average peak values is $y = kx + b$, among which, $y$ is the COD detection value of glucose solution, $x$ is the voltage average peak values, and $k$ and $b$ are constants. According to the linear fitting results, the relationship curve equation between the glucose standard solutions and the average peak values of chemiluminescence voltage is as follows:

$$y = 170.6x - 260.2 \tag{1}$$

In order to verify the detection accuracy of ozone chemiluminescence method, the COD values of the remaining four groups of solutions are calculated by Equation (1), and the calculation results are recorded in Table 4.

**Table 4.** Ozone chemiluminescence COD detection values of the remaining four groups of glucose solutions.

| Glucose Concentration (mg/L) | 9.37 | 28.12 | 32.8 | 46.86 |
|---|---|---|---|---|
| Voltage Average Peak Values (V) | 1.58 | 1.70 | 1.73 | 1.82 |
| COD Detection Values (mg/L) | 9.3 | 29.8 | 34.9 | 50.2 |

In order to observe the change trend and deviation between the COD detection value of the four groups of glucose solutions and the detection value of potassium dichromate method, the COD detection data of the two groups are further statistically sorted, as shown in Figure 14.

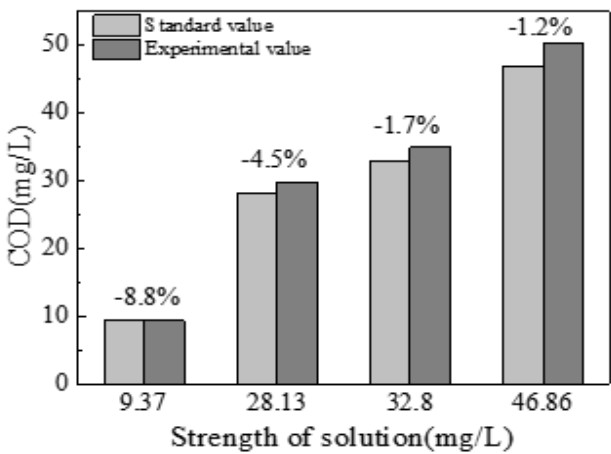

**Figure 14.** Comparison chart of standard and detection values of COD in four groups of glucose solutions.

Figure 14 describes the comparison between the COD test values of the potassium dichromic method for four groups of standard glucose solutions and the COD test values of this system and gives the deviation of the COD test values of each group of glucose solutions. It can be observed from the figure that the test value is generally slightly less than the standard value, and the average deviation of the COD detection value of the four groups of glucose solution is less than ±5% by calculation, indicating that the COD detection test system of ozone chemiluminescence has a good COD detection effect.

However, the deviation of detected values at the concentration of 9.37 mg/L glucose solution in the figure is greater than ±5%. This phenomenon may result from the systematic error brought by the current ozone chemiluminescence COD testing system. For example, the poor isolation effect of the camera obscura on the external magnetic field, noise and vibration result in signal interference, or ozone generator power instability results in ozone concentration differences. In the future, the existing COD detection system of ozone chemiluminescence should be further improved and optimized to further improve the accuracy of COD detection.

## 4. Conclusions

In this paper, a vortex T-type microfluidic detection chip was developed by microfluidic technology. According to the principle of ozone chemiluminescence, we designed a highly integrated COD detection system and carried out a quantitative test study on simulated water COD (glucose solution). The specific contents and results are as follows:

(1) Bubble formation tests in T-shaped microchannels were carried out. The results show that T-type microchannels have certain advantages for bubble formation. Therefore, a T-type microchannel can also be designed for a COD detection chip, which can generate ozone-based bubbles through flow shear action, and ozone equivalent per unit time can be effectively controlled by bubble encapsulation of ozone, in order to achieve full and effective reaction between ozone and organic matter and to produce the best intensity of chemiluminescence as well as enabling avoidance of excessive ozone caused by environmental pollution at the same time.

(2) The structural design and optimization of the COD detection chip were carried out. Based on the T-type channel, the micro-channel structure was designed as a T-type vortex structure with a width of 0.5 mm and a depth of 0.25 mm, and the luminous signal

detection test was carried out. The results showed that the chip structure is beneficial in the collection of chemiluminescence signals and can achieve a wide range of gas–liquid flow regulation; in addition, the structure of COD testing chips also provides a reference for the structural design of similar test chip samples.

(3) The experimental study of COD detection in a simulated water body was carried out. The test system was used to carry out 10 groups of COD tests of glucose solution, and the test value was compared with the standard value. The results showed that the average deviation of the two was less than ±5%, which verified the reliability and accuracy of the COD test system. Meanwhile, it has the advantages of a simple structure, short testing time, and no pollution, which provides a certain technical support for on-line real-time monitoring of water quality.

**Author Contributions:** Conceptualization, P.L. and Y.W.; methodology, B.X.; software, Y.W.; validation, P.L., Y.W. and B.X.; formal analysis, P.L.; investigation, Y.W.; resources, B.X.; data curation, Y.W.; writing—original draft preparation, Y.W.; writing—review and editing, P.L.; visualization, Y.W. All authors have read and agreed to the published version of the manuscript.

**Funding:** This work was supported by the Science and Technology Department of Jilin Province (20210203134SF, 20210203171SF, 20200301044RQ) and the "Thirteenth Five-Year" Science and Technology Project of Jilin Provincial Department of Education (JJKH20200036KJ).

**Data Availability Statement:** The data used to support the findings of this study are included within the article.

**Conflicts of Interest:** There are no conflict of interest regarding the publication of this paper.

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
