# Peer review of "Research on Micro-Quantitative Detection Technology of Simulated Waterbody COD Based on the Ozone Chemiluminescence Method"

_water, doi:10.3390/w14030328_

Round 1
Reviewer 1 Report
I missed the comparison of results to "true" COD methods, to both variants using potassium manganese or dichromate as oxidizing agent. It would be also very useful to compare this method with natural water not only with solution of glucose, because the solution of glucose is transparent but natural waters containing dissolved organic matter are more colourful and the organic matter itself could affect the luniniscence.
Author Response
Distinguished experts:
Hello!
Thank you very much for your hard work, and I'm really sorry for replying to you so late. We have modified the papers one by one according to your suggestions, and retained the modification traces by using the revision mode. In addition, the answers to each question can be found in the attachment.
If you have any questions, please feel free to contact us again. We look forward to receiving your reply!
Best wishes!
Yours sincerely.

Reviewer 2 Report
The article presents interesting research results and their analysis, concerning micro-quantitative Detection Technology of Waterbody COD. The introduction should be improved in terms of presenting the purpose of the research and the novelty of the proposed methodology, its advantages and the quality of test results. The description of the methodology and the presentation of the results should be extended.
There is no reference to the latest international literature. There are only 5 references from the last five years. The literature review and discussion should be significantly expanded. The discussion should be critical of the results presented in the literature, opposite the background of your own research results. Conclusions should be developed and detailed results of the conducted research should be presented.
Author Response

(The authors gave the same response as above.)

Reviewer 3 Report
Title: Research on Micro-Quantitative Detection Technology of Waterbody COD Based on the Ozone Chemiluminescence Method
T-type structural microchannel based on PDMS has advantages for forming micro-fine bubbles and detecting COD by chemiluminescence signal. The authors have used glucose concentration to carry out this experiment and predicted the accuracy with the standard potassium dichromate method. The reviewer found a good presentation and results. The weak point is that the authors have not used real wastewater to compare the results. The real water body contains many micro and macroparticles that may interfere with getting accurate results. If authors can perform a test, it will be a promising finding for the reader. The reviewer can recommend this paper for publication in this journal after a major revision.
##Comments on Abstract, Title, and References
- Please format the manuscript such as Abstract, Introduction, Experiment, Result and discussion, and conclusion.
- The abstract is informative. Authors can revise the title. Reviewer’s suggestion “Improved Chemical oxygen demand detection in water by Ozone Chemiluminescence Method.
- Please update references.
##Comments on Introduction
- Line 24-27: This hefty sentence should be split and clarify the meaning.
- Line 79-80: there should be space between numerical and unit. Please check entire the manuscript.
##Comments on detection principles and test environment
- Please discuss the unreacted ozone gas. Is there any option to collect the unreacted ozone from water? Fig. 6 represents the reaction is going on in the open condition. Is it okay?
- Table and Figure captions should be revised carefully entire the manuscript and avoid capitalizing each word.
- The detection of COD by K2Cr2O7 method shows a high value for Class II to Class V and inferior in Table 1. Why?
- Line 176: As can be seen from Table 2 that for 150mg/L glucose solution. I cannot find it. Please check.
- Is there any situation in that BOD can interfere with running the test of COD using said approach? Please include the statement in the manuscript.
- What will be the estimated cost to implement the method for practical use? Please include this section before the conclusion section.
- If possible, please perform a test using real wastewater and include the results in the manuscript.
Author Response

(The authors gave the same response as above.)

Round 2
Reviewer 1 Report
Accept
Author Response
Distinguished experts:
Hello!
Thank you again for your hard work. We have revised the paper according to your suggestions.
Wish you a happy life and smooth scientific research!
Best wishes!
Yours sincerely
Reviewer 2 Report
The authors introduced significant corrections and additions to the article, which improved its quality. In Fig. 5 the units should be marked on the axes of the plot. Numerous corrections can still be made to the description of the test results. I recommend accepting the article for publication after introducing minor corrections. Good luck!
Author Response
Dear Professor:
Greeting!
Thank you again for your hard work and recognition of our work. We have revised the paper according to your suggestion.
Wish you a happy life and smooth scientific research!
Best wishes!
Yours sincerely
Reviewer 3 Report
The Reviewer found the improvement of the manuscript and sound responses to the reviewer comments. The Reviewer recommends acceptance of the paper after a minor revision.
Minor correction:
1. Lines: 59-72 are in bold; and 2. Experimental will be in below.
2. Line 149: Figure 2 may be Figure 6.
3. Line 256: Figure 12 caption needs correction.
4. Line 276: Figure 11 may be Figure 13.
Author Response
Dear Professor:
Greeting!
Thank you again for your hard work and recognition of our work. We have revised the manuscript one by one as you suggested.
Wish you a happy life and smooth scientific research!
Best wishes!
Yours sincerely